# Contributions of mast cells and vasoactive products, leukotrienes and chymase, to dengue virus-induced vascular leakage

Ashley L St John[1,2]*, Abhay PS Rathore[3], Bhuvanakantham Raghavan[3], Mah-Lee Ng[3], Soman N Abraham[1,4]

[1]Program in Emerging Infectious Diseases, Duke-National University of Singapore, Singapore, Singapore; [2]Department of Pathology, Duke University, Durham, United States; [3]Department of Microbiology, National University of Singapore, Singapore, Singapore; [4]Departments of Pathology, Immunology, Molecular Genetics and Microbiology, Duke University, Durham, United States

**Abstract** Dengue Virus (DENV), a flavivirus spread by mosquito vectors, can cause vascular leakage and hemorrhaging. However, the processes that underlie increased vascular permeability and pathological plasma leakage during viral hemorrhagic fevers are largely unknown. Mast cells (MCs) are activated in vivo during DENV infection, and we show that this elevates systemic levels of their vasoactive products, including chymase, and promotes vascular leakage. Treatment of infected animals with MC-stabilizing drugs or a leukotriene receptor antagonist restores vascular integrity during experimental DENV infection. Validation of these findings using human clinical samples revealed a direct correlation between MC activation and DENV disease severity. In humans, the MC-specific product, chymase, is a predictive biomarker distinguishing dengue fever (DF) and dengue hemorrhagic fever (DHF). Additionally, our findings reveal MCs as potential therapeutic targets to prevent DENV-induced vasculopathy, suggesting MC-stabilizing drugs should be evaluated for their effectiveness in improving disease outcomes during viral hemorrhagic fevers.

**\*For correspondence:**
ashley.st.john@duke.edu

**Competing interests:** The authors declare that no competing interests exist.

**Reviewing editor**: Ruslan Medzhitov, Yale University, United States

## Introduction

Millions of individuals are infected yearly with DENV and some of these develop potentially deadly disease states, such as DHF and Dengue Shock Syndrome (DSS), both of which involve increases in vascular permeability, plasma leakage into tissues and hemorrhaging within internal organs. In severe cases, circulatory failure and death can occur (*Halstead, 2007*). Currently, no targeted treatments exist to stabilize the vasculature during severe DENV complications, in part due to our lack of understanding of the mechanisms of DENV-induced vascular leakage (*St John et al., 2013*).

MCs are well-established cellular regulators of vascular integrity, tone and function. They line blood vessels and produce many vasoactive mediators that have redundant functions in inducing vascular permeability. Some of these are pre-stored and can act nearly instantaneously on vascular endothelium, including TNF, proteases (e.g., chymase and tryptase), and heparin (*Brett et al., 1989*; *Huang et al., 1997*; *Sendo et al., 2003*; *Oschatz et al., 2011*). Other de novo synthesized vasoactive factors include leukotrienes, prostaglandins, VEGF and, again, TNF (*Flower et al., 1976*; *Dahlen et al., 1981*; *Sendo et al., 2003*). With their activation, MC-derived factors act in concert to promote the break down of junctions between endothelial cells, plasma leakage and edema within tissues, as well as to reduce clot formation and increase blood flow in the vicinity of MC-activated endothelium (*Kunder et al., 2011*). MCs are also versatile detectors of infection that respond strongly to direct activation by DENV (*St John et al., 2011*) as well as indirect, antibody-mediated activation (*Sanchez et al., 1986*). We also

**eLife digest** Dengue fever is an infectious tropical disease that is transmitted by mosquitoes carrying dengue virus. Almost half of the world's population lives in dengue-plagued regions and it is estimated that between 50 and 100 million people are infected annually. However, there is no effective vaccine and researchers have only a limited understanding of the mechanisms behind the disease.

While most infected individuals experience fever, headache, muscle and joint pain, and a skin rash, a small percentage go on to develop a life-threatening condition known as dengue hemorrhagic fever (DHF). This is marked by internal bleeding, and by the leakage of water and salts from blood vessels (vascular leakage). At present, it is difficult to tell which patients will develop this complication, making it hard to tailor treatment appropriately.

Here, St John et al. reveal that the vascular leakage that occurs in DHF is triggered by mast cells, which line blood vessels and regulate their permeability through the release of molecules such as histamines and leukotrienes. They also found that mice deficient in mast cells did not show dengue-induced vascular leakage, and that wild-type animals treated with drugs that block the actions of proteins produced by these cells, showed less vascular leakage than controls. Moreover, levels of an enzyme called chymase, another mast cell product, are higher in human patients with DHF than in those with dengue fever. Since chymase release occurs early in infection, tests for the presence of this enzyme could be used to predict which patients are likely to develop DHF.

The work of St John et al. suggests new lines of inquiry into the mechanisms that lead some individuals infected with dengue fever to develop DHF, and indicates that drugs that target mast cells could offer an effective treatment.

have recently reported that the degranulation response of MCs to DENV does not require active viral replication within MCs, since UV-inactivated virus was sufficient to promote MC degranulation (*St John et al., 2011*).

During localized inflammation in vivo, MC-induced vascular permeability promotes the delivery of humoral factors into a tissue and recruitment of immune cells, such as NK cells and T cells to the sites of DENV infection (*St John et al., 2011*). However, systemic or aberrant activation of MCs is a contributing factor to many pathological conditions associated with leakage of blood vessels, including anaphylaxis, asthma, aneurysm, and others (*Frangogiannis et al., 1998*; *Williams and Galli, 2000*; *Finkelman, 2007*). Severe DENV outcomes in human patients have been associated with high levels of vasoactive factors that MCs produce (*Vitarana et al., 1991*; *Tseng et al., 2005*), high levels of products that enhance MC responses, including IgE (*Koraka et al., 2003*; *Mabalirajan et al., 2005*), and MC-activation associated symptoms, such as rash and thrombocytopenia (*World Health Organization and the Special Programme for Research and Training in Tropical Diseases, 2009*). Together, these associations prompted us to examine the role of MC in DENV-induced vasculopathy. We began this work with the hypothesis that MCs modulate the vascular endothelium during DENV infection with pathological consequences.

## Results

### Observation of MC activation and vascular leakage in DENV-infected tissues

First, we sought to visually examine the interactions between MCs and blood vessels in the presence of DENV. The mouse ear tissues presented in *Figure 1A–C* were prepared for transmission electron microscopy (TEM) 24 hr after sub-cutaneous injection of saline (*Figure 1A*) or $1 \times 10^5$ PFU of a clinical isolate of DENV (*Figure 1B–C*). This inoculating dose was used since it approximates our understanding of the infective dose instilled into the skin by mosquito injection (*Bente and Rico-Hesse, 2006*). Control, saline-injected tissues contained apparently unactivated MCs, located proximal to intact blood vessels, which were densely packed with intra-cellular granules. Several changes to the ultrastructure of DENV-injected tissues were noted, contrasting the quiescence of saline-injected tissues. DENV-injected tissue contained many activated MC located near vessels (*Figure 1B*). These could be identified

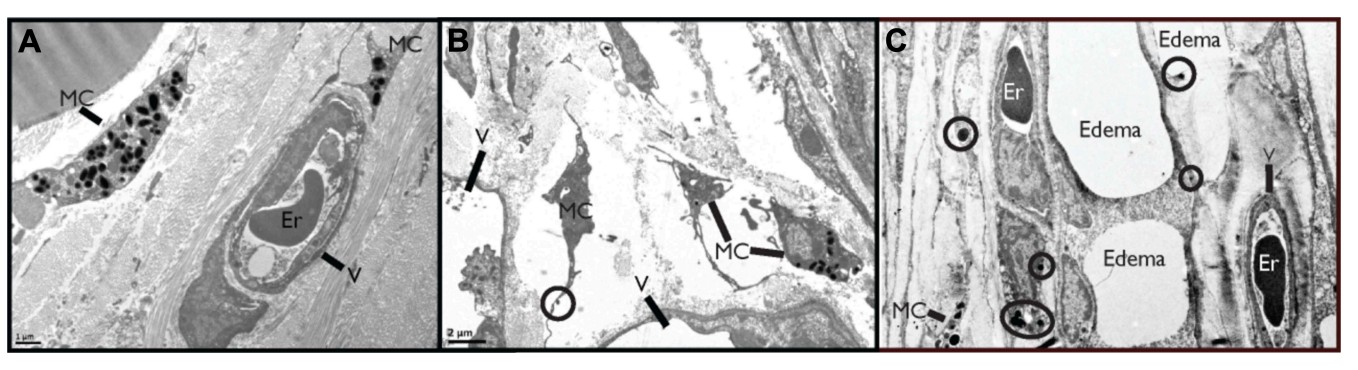

**Figure 1**. DENV-Induced MC activation and microstructural changes surrounding blood vessels. TEMs acquired from mouse ear tissue 24 hr after either saline injection (**A**) or 1 × 10⁵ PFU of DENV (**B**) and (**C**). Figure labels for (**A**)–(**C**): MC: mast cell; V: vessel; Er: erythrocyte. (**A**) In control tissues, apparently quiescent, granulated, MCs can be visualized in proximity to a blood vessel. (**B**) A field containing many MCs that appear activated due to their reduced granularity and cytoplasmic projections that are characteristic of MC degranulation. A granule that is being released is circled. This image also contains portions of two vessels, visible on the left and bottom sides. (**C**) MCs are closely associated with vessels in tissue that shows signs of fluid pooling or edema. Extracellular granules are visible throughout the tissue and are circled. A blood vessel containing an erythrocyte is labeled on the right side of the image and a second vessel, likely a lymphatic vessel, is visible on the left side.

by the characteristic lengthened cellular protrusions that remain after degranulation and the presence of residual intracellular granules. Occasionally granules could be identified in the process of being released (circled in *Figure 1B*). In addition to MC degranulation, evidence of fluid pooling or edema could be observed in infected tissues, also in close proximity to MCs, extracellular MC granules, and vascular endothelium (*Figure 1B*). These images are representative of our initial observation that MC activation and loss of vascular structure appear intimately associated in DENV-injected tissues and they led us to undertake further studies to examine the role of MCs in DENV-induced vascular leakage.

## Characterization of systemic DENV infection in the WT mouse model

Prior to experimentally addressing the role of MC responses to immune pathology and vascular leakage on a systemic level, we undertook studies to validate the use of an immunocompetent animal model that could provide measurable evidence of vascular changes. For DENV, a prominent portion of the literature raises doubt as to whether wild type (WT), immunologically intact, mice can be productively infected by DENV or are valid hosts for undertaking studies to investigate the mechanisms of vascular permeability (*Yauch and Shresta, 2008*). In spite of this, studies to address the role of MC-promoted inflammatory responses to DENV require the context of a host that is able to generate normal immune responses, whether productive or pathological (*St John et al., 2013*). Although WT mice have been shown to be less susceptible to infection than mice lacking the capacity of innate immune activation (*Shresta et al., 2004*), we, like others (*Boonpucknavig et al., 1981*; *Huang et al., 2000*; *Atrasheuskaya et al., 2003*; *Paes et al., 2005*; *Chen et al., 2007*), observe and have previously reported WT mice can sustain replicating DENV infection (*St John et al., 2011*). Others have also used WT mice to examine DENV-induced vascular hemorrhaging or pathology (*Chen et al., 2007*; *Assuncao-Miranda et al., 2010*). Based on these multiple lines of evidence supporting the feasibility of undertaking studies in the WT mouse model, we optimized a system of infecting WT mice to generate systemic DENV infection, by injecting 1 × 10⁶ plaque-forming units (PFU) of DENV, intra-peritoneally (i.p.). This route of infection was chosen in order to bypass the stages of natural, peripheral infection when WT mice can also clear virus quickly. Cohorts of infected WT mice were monitored for evidence of vascular events consistent with DENV symptoms (*Figure 2A,B*) and for evidence of productive DENV replication (*Figure 2C,D*) for 5 days after the initiation of systemic infection. Two measures of vascular function that are associated with DENV pathology in human patients are thrombocytopenia and elevated hematocrit values, the latter of which is a key test used to diagnose DENV mild to moderate vascular leakage or hemorrhaging and vascular pathology clinically, although severe hemorrhaging can be accompanied by suddenly lowered hematocrit values (*World Health Organization and the Special Programme for Research and Training in Tropical Diseases, 2009*). Since hematocrit measures the packed red blood cell

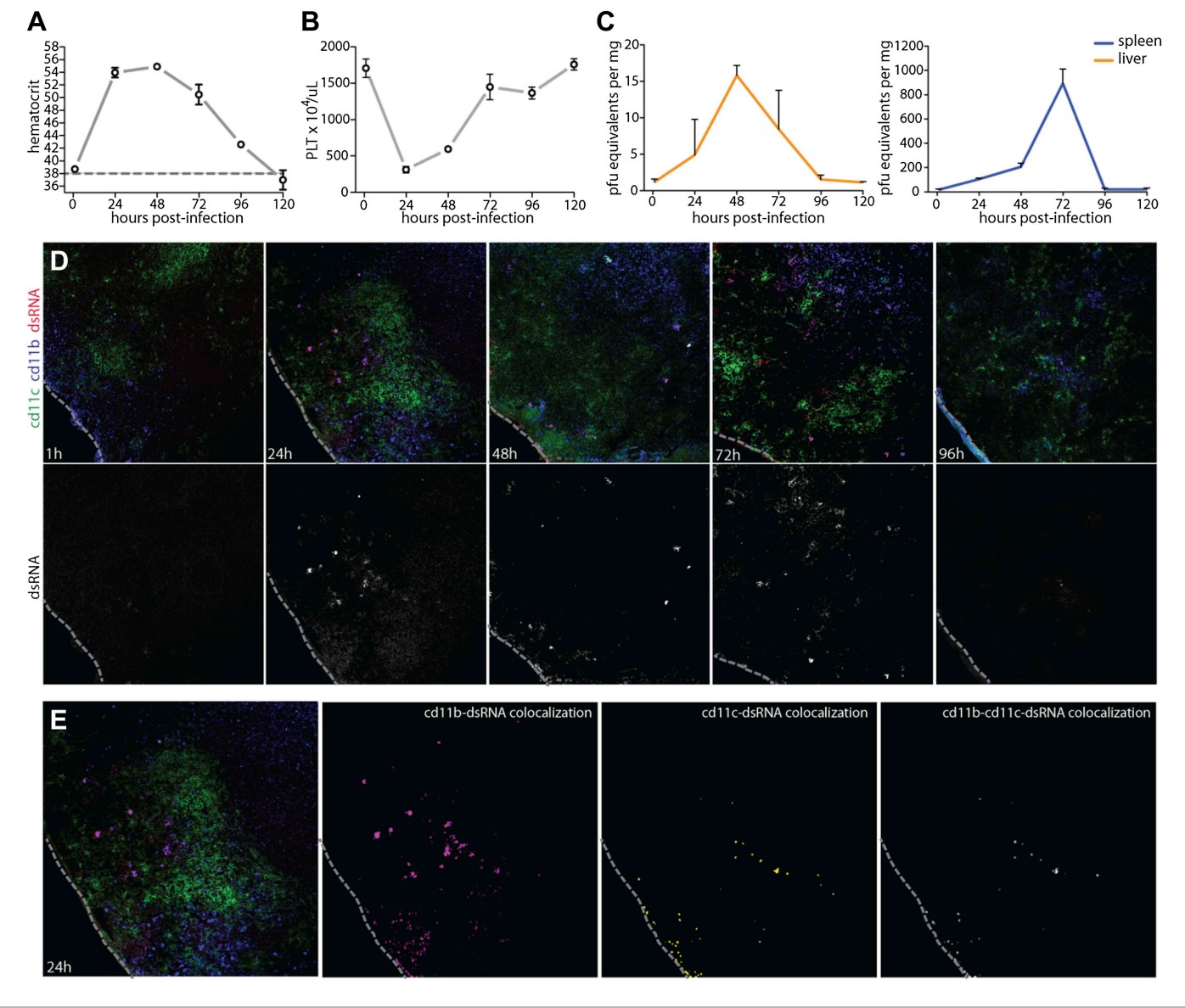

**Figure 2**. Vascular pathology and DENV replication in the WT mouse model. Cohorts of mice were monitored for 5 days following the experimental establishment of systemic infection using DENV clinical isolate Eden2, by intra-peritoneally. injection. Blood was obtained at 1 hr after infection and at subsequent 24 hr time points then analyzed to determine the (**A**) hematocrit values and (**B**) platelet concentration for individual infected mice (n = 3–4 at each time point). A dashed line represents the average baseline values for uninfected, naïve animals. (**C**) At the same time points, tissue was harvested from the spleen and liver and real time PCR was performed after cDNA conversion in order to quantitate DENV genome copies (PFU equivalents) in the tissue, which was normalized to the tissue mass. Error bars for panels (**A**–**C**) represent the SEM and are included for all groups; where they are not visible, variation was small. (**D**) Images of spleen sections along this time course are presented, where 10-µm-thick sections were stained for DENV replication (dsRNA, red), monocytes and/or macrophages (cd11b, blue), and dendritic cells (cd11c, green). Bottom panels in (**D**) show the isolated dsRNA panel. (**E**) Co-localization images were generated using ImageJ software and reveal that dsRNA staining co-localized predominantly with the monocyte/macrophage marker cd11b, and to a lesser extent, the DC marker, cd11c. For panels (**D** and **E**), the dashed line denotes the border of the spleen in the tissue section.

volume as a percentage of the total blood volume, it is a measure of plasma loss from the circulation. WT mice experimentally infected with this DENV clinical isolate experience a rise in hematocrit values that peaks 24–48 hr after infection and remains significantly increased over naive mice for 4 days post-infection (*Figure 2A*), demonstrating that measureable evidence increased vascular permeability can be detected in the WT mouse model over a prolonged time course. In line with additional symptoms of DENV infection in humans, WT mice also experienced a drop in platelets over the first 48 hr of

infection before recovering by day 3 (*Figure 2B*). Thus, with regards to quantitative measures of vascular pathology, WT mice display multiple symptoms consistent with human DENV during infection.

To further characterize the infection profile of this WT model we also examined the liver and spleen for evidence of viral replication, since these are DENV-target organs in humans (*Rosen et al., 1989*). By real time PCR for DENV NS1, we were able to determine that viral replication peaked in the liver at 48 hr and in the spleen at 72 hr (*Figure 2C*). This evidence of productive DENV infection was supported by a secondary technique of immunofluorescence staining for the replication intermediate, double-stranded RNA (dsRNA). As expected, dsRNA staining could not be observed only 1 hr after infection, prior to when systemic replication could have been initiated on a detectable scale (*Figure 2D*). However, dsRNA staining could be observed in the spleen beginning at the 24 hr time point and this staining remained apparent throughout the time course, although greatly visually reduced by day 4 post-infection (*Figure 2D*), consistent with real time PCR quantification of splenic viral copies (*Figure 2C*). Staining for dsRNA also co-localized with markers for monocytes or macrophages (cd11b$^+$cd11c$^-$) and dendritic cells (cd11b$^±$cd11c$^+$) (*Figure 2E*), and these are well-established DENV-target cell types (*Wu et al., 2000*; *Tassaneetrithep et al., 2003*; *Jessie et al., 2004*; *Blackley et al., 2007*). This supports that the DENV clinical isolate, Eden2 is able to replicate in WT mice, with an infection profile consistent with what is known about human DENV infection, in terms of target organs and cell types.

## MC-dependent vascular permeability during DENV-Infection

Having observed that DENV-induced MC activation is linked to microstructural vascular changes (*Figure 1*) and having validated our model of systemic DENV infection (*Figure 2*), we undertook studies experimentally to assess the contributions of MCs to DENV-induced vascular pathology. To reveal any contributions of MC to vascular leakage during DENV infection, we compared the responses of immunologically intact WT hosts to mice lacking MCs (Sash). Sash mice are a well-established MC-deficient model, where MCs are almost entirely absent due to a key point mutation in the promoter of the *ckit* gene that predominately affects its expression within the MC lineage (*Grimbaldeston et al., 2005*). To determine if DENV-infected WT mice can display systemic levels of MC activation, we measured the MC-specific product, MCPT1, in mouse serum by ELISA in naïve and DENV-infected mice. After intra-peritoneally. injection of DENV as described above, MCPT1 was detectable in the serum of WT mice at 24 hr and continuing beyond 48 hr (*Figure 3A*). As expected, Sash mice had undetectable serum levels of MCPT1 at baseline, as did uninfected congenic wild-type (WT) controls (*Figure 3A*). Having verified that our infection model allows systemic detection of a key MC-derived vasoactive product, we next examined readouts of vascular leakage in this system by hematocrit analysis, again, since this is a key measure of both mild and severe vascular leakage with clinical relevance. The results of hematocrit analysis supported a MC-dependency to vascular leakage since MC-sufficient WT animals had elevated hematocrit values with DENV infection, but Sash mice did not (*Figure 3B*). Importantly, since a few other processes (such as melanocyte migration) are affected in Sash mice (*Grimbaldeston et al., 2005*), repletion studies were performed in these experiments as a control. To replete MCs in order to establish that MCs are sufficient to restore the WT functional phenotype, bone marrow-derived MCs (BMMCs) were passively infused intra-venously. Since repletion of Sash mice with MCs, alone, was able to restore elevated vascular leakage (*Figure 3B*), the DENV-induced affect of increased vascular permeability appears to be MC dependent.

To provide a second measure of vascular leakage supporting the hematocrit data, we undertook studies to examine the MC-dependency of dye leakage into tissues during DENV infection, using the dye Evans blue. Evans blue leakage studies are commonly used to assess models of viral hemorrhaging (*Gowen et al., 2010*), however, we modified the standard technique so as to improve the sensitivity to detect sub-hemorrhagic vascular leakage by flushing excess dye from the vascular system by saline perfusion, prior to quantitation. Following this strategy, mice were injected by tail vein with the tracking dye, Evans blue, 30 min prior to euthanasia at a 24-hr time point after infection. Initial examination of the internal organs of mice did not reveal visible leakage of dye (*Figure 3—figure supplement 1*), which is consistent with the observations by others that immunocompetent WT mice do not always exhibit gross pathological vascular leakage with DENV infection (*Raut et al., 1996*). However, when mice were perfused with saline, increased in vascular permeability was visible in infected mice due to residual blood and dye remaining in the tissues (*Figure 3—figure supplement 1*). Therefore, this model allows us to detect the earliest signs of vascular leakage quantitatively and to assess the effects of MC detection of DENV to vascular integrity in vivo.

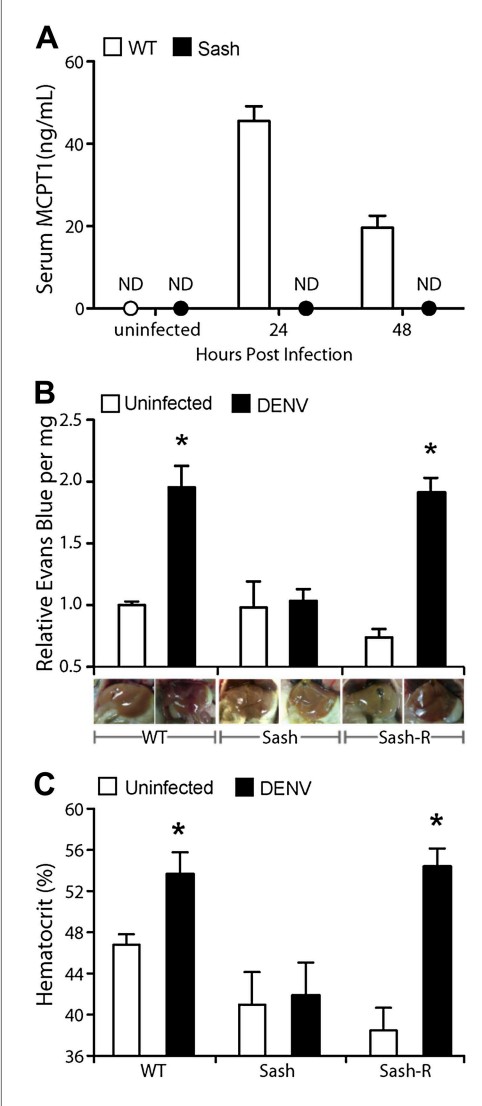

**Figure 3**. Vascular leakage during DENV infection is MC-dependent. (**A**) Graph depicts the serum concentration of MCPT1, which was quantified using serum obtained from WT or Sash mice, 24 and 48 hr after intra-peritoneally. injection with $1 \times 10^6$ PFU of DENV. MCPT1 was not detected (ND) in uninfected WT mice and uninfected or infected Sash mice. Error bars represent the SEM of ELISA replicates using pooled serum samples from n = 4 animals. To compare vascular leakage in infected vs uninfected WT, Sash, and Sash-R mice, (**B**) hematocrit analysis using heparinized blood and (**C**) quantitation of Evans blue dye leakage into liver tissue were performed 24 hr after infection with DENV. Representative images of mouse livers after saline perfusion are presented below the respective data bars to support that visually perceivable vascular leakage occurred in DENV-infected animals. For (**B** and **C**), error bars represent the SEM where values were obtained from individual infected mice n = 3–6 per group. * indicates a significant increase over uninfected controls; p≤0.05.

*Figure 3. Continued on next page*

Using this Evans blue perfusion model, we determined the extent of vascular leakage induced in 24 hr after intra-peritoneally instillation of $1 \times 10^6$ plaque-forming units (PFU) of DENV in WT and Sash mice. Liver tissue was harvested and homogenized to obtain quantitative values regarding the amount of dye leakage into the tissue and also visually examined for evidence of leakage (*Figure 3C*). After perfusion, WT and Sash-R mice showed visually apparent increases in vascular permeability in the liver that were supported by Evans blue quantitation, however, Sash mice did not (*Figure 3C*). Additional highly vascularized organs, such as the kidney, also showed visually perceivable evidence of MC-dependent leakage of blood into the tissues that was not apparent in Sash mice after saline perfusion (*Figure 3—figure supplement 2*). Since the liver of mice contains very few MCs compared to other species such as humans (*Bois et al., 1964*; *Koda et al., 2000*), these findings suggested that systemic MC effects were likely responsible for the observed vascular changes. MC-dependent vascular leakage was also directly correlated to increased MC-derived product MCPT1 (*Figure 3A*). Together, these findings, combined with the data obtained by hematocrit analysis (*Figure 3B*) point to a pivotal role for MCs in promoting DENV-induced increases in vascular permeability.

## Targeting of MCs and their products to reduce vascular pathology

Due to the contributions of MCs to many pathological inflammatory disorders, there are a number of available drugs that target MCs or products that they produce after activation. Informed by our studies suggesting that MCs augment DENV-induced vascular leakage in animal models (*Figure 3*), we attempted to block this vascular leakage using a panel of these drugs. We chose two well-established MC-stabilizing compounds that are clinically available, cromolyn and ketotifen (*Theoharides et al., 1980*; *McClean et al., 1989*), and two treatments that block MC products that are known to promote vascular leakage, an anti-TNF blocking antibody and montelukast, a leukotriene receptor antagonist (*Busse et al., 1999*). None of these drugs act exclusively on MCs or MC-derived products but they are known to modulate the functions of MCs effectively in a clinical context. Drugs were administered once, 30 min after intra-peritoneally instillation of $1 \times 10^6$ DENV and vascular leakage was assessed 1 day after infection. Both MC-stabilizing compounds significantly decreased vascular leakage compared

*Figure 3. Continued*

The following figure supplements are available for figure 3:

**Figure supplement 1**. Representative images of livers from uninfected mice, or mice infected with $1 \times 10^6$ PFU of DENV.

**Figure supplement 2**. Representative images of the kidneys of WT mice or Sash mice that were infected with $1 \times 10^6$ PFU of DENV.

to untreated DENV alone, quantitated by Evans blue leakage as well as hematocrit analysis (*Figure 4A*). Blocking TNF has previously been shown to be effective in limiting vascular leakage at late time points of infection in immunocompromised mice lacking Type I and II Interferon receptors (IFN-α,β,γ-R$^{-/-}$, or AG129), which is a mouse model of DENV viremia (*Zellweger et al., 2010*). In our WT mouse model, when TNF-blocking antibodies were administered 1 hr after systemic infection was initiated by intra-peritoneally injection, the average Evans blue dye leakage and hematocrit values were lower; however, this did not reach statistical significance (*Figure 4A*). Blocking leukotriene function using the drug montelukast also allowed striking, significant reductions in dye leakage compared to untreated mice (*Figure 4A*), suggesting leukotrienes can also contribute to DENV-induced vascular permeability. Although cromolyn's inhibitory influence on MCs is well established (*Theoharides et al., 1980*), to support a direct effect of the MC-stabilization strategy on MCs in this DENV infection model, we again compared serum MCPT1 in untreated and cromolyn-treated DENV-infected mice. As expected, levels of MCPT1 were reduced by cromolyn treatment at 24 hr, and by 48 hr, serum MCPT1 was undetectable in treated mice, yet still elevated in untreated mice (*Figure 4B*). By 24 hr, there was a trend towards a slight increase in the average number of DENV genome copies in the serum, but this did not reach levels of statistical significance (*Figure 4C*). Since three drugs from this panel, cromolyn, ketotifen and montelukast, all were successful in restoring both the levels of dye detection in the liver tissue and hematocrit values to levels that were not statistically different from baseline levels (*Figure 4A*), these data highlight the potential of MCs to serve as therapeutic targets to limit DENV pathology.

Additional cell types can also produce some of the vasoactive products that MCs produce, including cytokines and eiocosinoid mediators. To determine if DENV elicits products from MCs, directly, that act on vascular endothelial cells, we performed trans-well assays to measure the responses of mouse endothelial cell (EOMA) monolayers to MC-derived products. Supernatants from DENV activated or control mast cells were transferred to monolayers of EOMA cells and the trans-endothelial resistance (TER), a measure of the barrier function of a cell monolayer that decreases as permeability increases, was measured after 24 hr to determine if DENV-activated MCs specifically promote loss of barrier function. We observed a significant drop in the TER reading relative to the baseline for EOMA cells exposed to DENV-activated MC supernatant but not to unactivated MCs or virus alone (*Figure 4D*). Cromolyn treatment of MCs also prevented this reduced barrier function, as did both blockade of leukotrienes with montelukast or inhibition of chymase with a cocktail of specific inhibitors including chymostatin and soybean trypsin inhibitor (*Figure 4D*). To confirm the role of MC-derived leukotrienes in promoting endothelial cell permeability in an additional experimental system, we tested whether leukotriene-deficient MCs are able to induce comparable vascular permeability to WT MCs. For this, we purified MCs from the peritoneal and pleural cavities of mice that lack 5-lipoxygenase (5-LO-KO), the enzyme required for leukotriene synthesis for use in trans-well assays. For comparison, MCs were also purified from WT and TNF-KO mice. Purified MCs were stained to verify that the isolated cells demonstrated a granulated MC morphology (*Figure 4E*, inset). We observed a significant reduction in the TER relative to baseline in EOMA cells exposed to supernatants from DENV-activated WT or TNF-KO MCs that did not occur in EOMA cells exposed to supernatant from DENV activated 5-LO-KO MCs (*Figure 4E*). 5-LO appears to mediate direct activity of DENV-activated MCs on endothelial cells since DENV-exposed WT MCs have a significantly greater impact on the TER of EOMA cells compared to MCs from 5-LO-KO mice. These data also provide evidence that MCs products have the potential to be direct intermediaries in promoting vascular permeability of endothelial cells that is not induced by virus alone treatment (*Figure 4D,E*).

To give our findings context in another established DENV mouse model, we also undertook experiments in IFN-α,β,γ-R$^{-/-}$ mice, since these mice can sustain replicating DENV infection for many days (*Zellweger et al., 2010*) and are a common model used for testing DENV anti-viral drugs (*Rathore et al., 2011*). *Figure 5A* contains images of the mouse mesentery (the membrane that anchors the gut within the peritoneal cavity), stained for blood vessels, MCs and DENV replication, 24 hr after intra-peritoneally

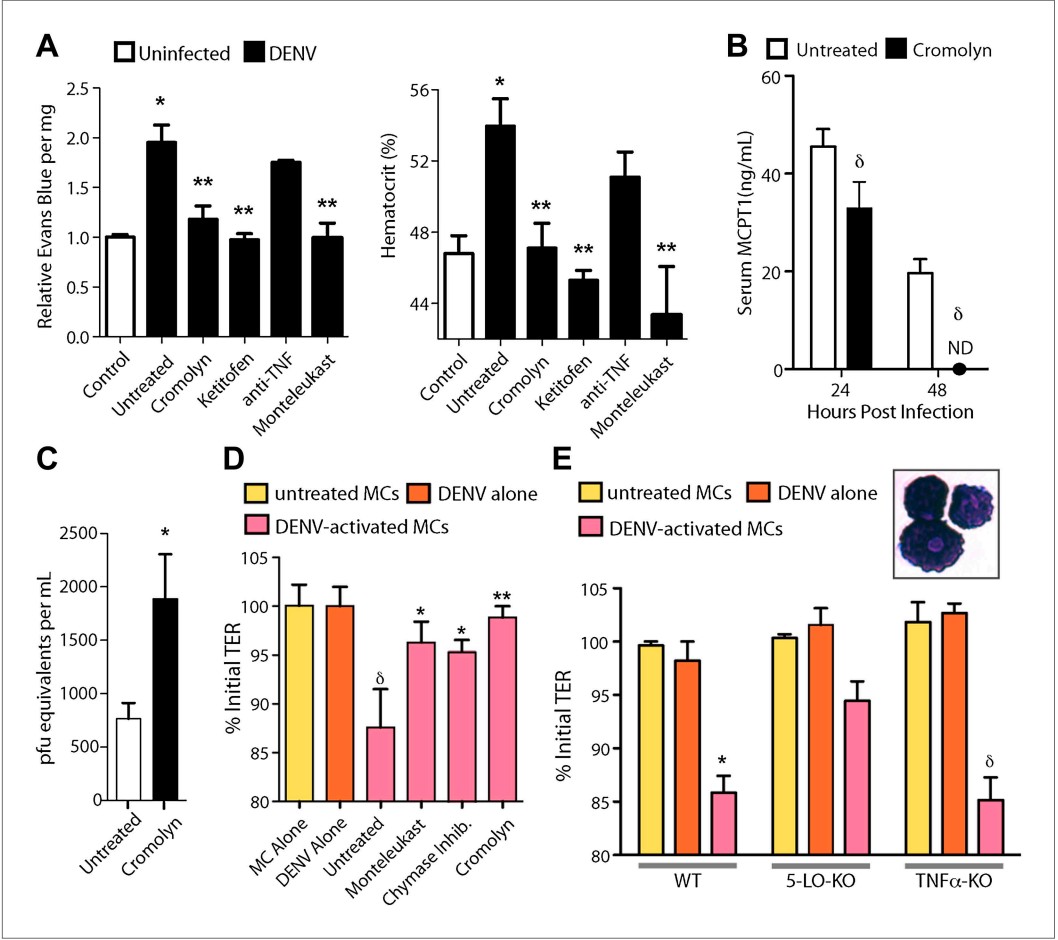

**Figure 4**. Drugs targeting MCs and their products improve DENV-induced vasculopathy. (**A**) Evans blue dye perfusion studies and hematocrit analysis were performed to determine the vascular leakage in mice infected intra-peritoneally with $1 \times 10^6$ PFU of DENV. Serum was obtained from uninfected mice, DENV-infected and untreated mice, and mice that received MC-stabilizing or MC-product targeting treatments (see 'Materials and methods') 24 hr after infection. Error bars represent the SEM of values obtained from individual animals n = 3–6 per group. Data was analyzed by ANOVA with Bonferroni post-tests to determine significance; * indicates a significant increase over control (uninfected) values and ** indicates a significant decrease from DENV-infected, untreated values; $p \leq 0.05$. (**B**) Serum ELISA for MCPT1 was performed using pooled serum from DENV-infected, untreated mice and DENV-infected, cromolyn-treated mice. Significance was determined by ANOVA; δ indicates a significant decrease compared to untreated controls; $p \leq 0.05$. (**C**) Viral genome copies were quantified in the serum of mice infected with DENV that were either untreated or treated with cromolyn. The moderate increase with cromolyn treatment was not significant with p=0.09 and n = 5. (**D**) Trans-well assays demonstrate the direct activity of MCs and MC products on permeability of a monolayer of EOMA cells. Significance was determined by ANOVA. δ indicates a significant decrease in TER compared to exposure to supernatants from untreated MC or DENV alone treatment (p<0.05). Groups treated with montelukast or chymase inhibitor cocktail significantly increased TER over untreated EOMA cells exposed to supernatant from DENV activated MCs; *p<0.05. Cromolyn treatment during DENV exposure resulted in increased TER over supernatants from untreated DENV-exposed BMMCs **p<0.01. (**E**) Trans well assays were also performed using peritoneal and pleural cavity MCs isolated by antibody labeling and magnetic separation. Purified MCs, which have abundant eosinophilic cytoplasmic granules, are imaged in the inset. Purified MCs from WT, 5-LO-KO, or TNF-KO mice were untreated or treated with DENV (MOI = 5) for 1 hr prior to isolation of supernatant for exposure to EOMA cells. Supernatants from both WT and TNF-KO MCs resulted in a significant reduction in the TER of EOMA cells with exposure compared to controls, determined by ANOVA; for δ p<0.05. 5-LO-KO showed a trend towards slightly reduced TER, but this was not significant since p=0.06. DENV activated WT MCs promoted significantly reduced relative TER readings compared to DENV activated 5-LO-KO MCs, determined by t-test *p=0.01. Similar results were obtained in a second independent vascular endothelial cell line, SVEC4-10EHR1 (***Figure 4—figure supplement 1***).

*Figure 4. Continued on next page*

*Figure 4. Continued*

The following figure supplements are available for figure 4:

**Figure supplement 1**. Supernatants from both WT and TNF-KO MCs resulted in a significant reduction in the TER of monolayers of vascular endothelial cell line SVEC4-10EHR1 with exposure compared to controls, determined by ANOVA; for δ p<0.05.

injection of $2 \times 10^5$ DENV. In control tissues, blood vessels appeared intact and were surrounded by tightly granulated MCs (*Figure 5A*). However, blood vessels during DENV infection appeared to have lost structural integrity and were surrounded by tiny avidin-staining particles, signifying extensive MC-degranulation within the vicinity of the vessel (*Figure 5A*). It was apparent that blood vessels in DENV-infected tissues also have decreased staining for the marker of vascular endothelium, CD31 (or PECAM), which functions as an adhesion molecule within endothelial junctions (*Figure 5A*). Proximal DENV-infected cells were revealed through staining for the DENV non-structural protein 3 (NS3) (*Figure 5A*), which is produced only during viral replication. Thus, evidence of MC activation and vascular leakage were also observed in these mice.

In this system, we delayed treatment with cromolyn until 1 day after systemic infection was initiated via intra-peritoneally injection, as performed previously. When blood was collected 3 days after infection and after two daily doses of cromolyn, we again observed that hematocrit values were reduced significantly compared to DENV-infected, untreated mice (*Figure 5B*). This apparent decrease in vasculopathy with cromolyn treatment even occurred in spite of an approximate doubling in the serum viral titer (*Figure 5B*). Although cromolyn has additional immunosuppressive functions that are not MC-restricted and these data were obtained in a mouse model having severe defects in anti-viral responses, they appear to emphasize the dual contributions of MCs to immune protection and pathology during DENV infection.

## DHF in humans is preceded by heightened MC activation

After obtaining data from two separate mouse models implicating MCs in the mechanism of DENV vasculopathy, as well as evidence that drugs that target MCs or their products can restore vascular integrity, we sought to validate this key finding in humans. To determine if MC activation occurred in human DENV patients, we tested serum from patients for the MC-specific product, chymase, by ELISA (*Figure 6A*). This human homologue of mouse MCPT1 is also known to increase vascular leakage over a prolonged time course (*He and Walls, 1998*). All sera were obtained from a previously described DENV clinical study (*Low et al., 2006*; *Fink et al., 2007*), where early serum samples were obtained during acute DENV infection (1–3 days after onset of fever), followed by a second sample during the defervescent stage of infection (4–7 days after onset of fever). The sera tested fell into the following groups: (1) control healthy human sera, (2) control sera from individuals with fever that were referred to the study as possible DENV cases but were negative for DENV by laboratory tests employing RT-PCR for DENV RNA, (3) sera from patients that were positive for DF by clinical diagnosis as well as RT-PCR and (4) sera from patients that were DENV positive by clinical diagnosis, molecular tests, and were also diagnosed during the study to have DHF, rather than the less severe DF. Strikingly, we found that those patients that were diagnosed with DF or DHF showed chymase levels in serum obtained during the acute phase of infection were significantly higher than levels in the serum of either healthy controls or individuals with fever that were DENV negative by RT-PCR (*Figure 6A*). This trend of heightened levels of chymase in the blood of DF and DHF patients persisted to the second time point of blood collection (*Figure 6B*), illustrating a prolonged course of elevated MC products occurs during infection. During the acute phase of infection, DF patients displayed an increase in serum chymase that was approximately 10 times higher than to healthy individuals or DENV-negative patients while, in DHF patients, chymase levels 30 times higher than healthy controls were detected (*Figure 6C*). Heightened levels of serum chymase were also maintained in DF and DHF patients relative to healthy controls during the defervescent stage of infection (*Figure 6C*; *Figure 6—figure supplement 1*). Since there is an approximate threefold increase in the levels of serum chymase in the blood of DHF patients compared to DF patients, this finding also illustrates a correlation between the levels of chymase and the severity of vasculopathy experienced in DENV-infected humans (*Figure 6A–C*). Patient serum samples were DENV serotypes 1-, 2-, or 3-positive, and the infecting serotype also appeared to be a significant, but minor influence to levels of chymase detected (*Figure 6D*). For example, chymase concentrations

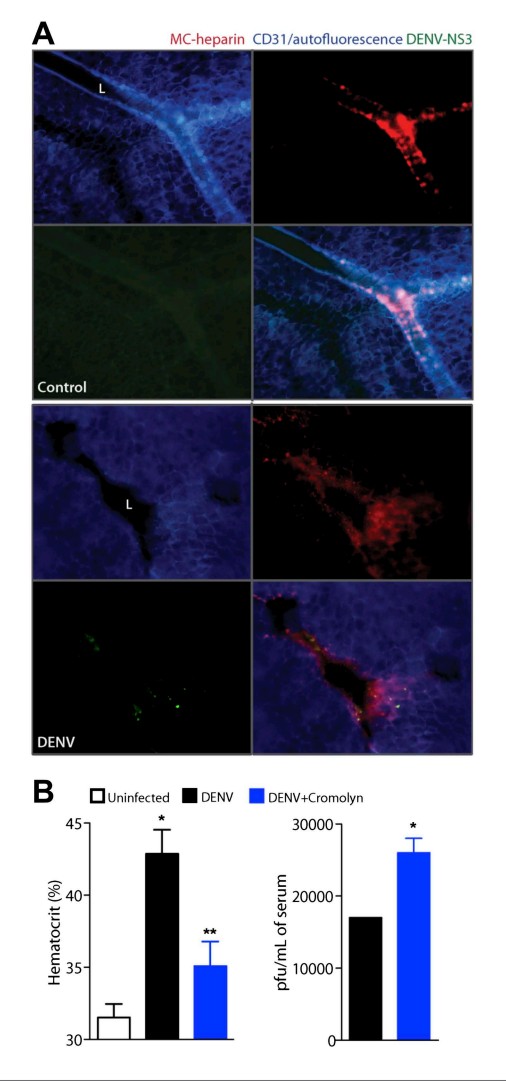

**Figure 5**. Cromolyn is effective in the IFN-α,β,γ-deficient mouse model to limit DENV-induced vasculopathy. (**A**) Images are presented for control (top) and DENV-infected mesentery tissue (bottom) in channel series showing staining for blood vessels (CD31, blue), MC granules (MC-heparin by probing for Avidin, red), and viral replication (NS3, green), as well as the merged image. Mesentery tissue from the DENV-permissive mouse strain, AG129, was isolated from control or DENV-infected tissue at 24 hr after intra-peritoneally injection of $2 \times 10^5$ PFU of DENV strain Eden2, followed by immunostaining in whole mount and viewing at 20× magnification. MCs can be observed lining the blood vessels (branches of the mesenteric artery) in control tissue (left). Discrete avidin-staining particles suggest extensive degranulation in DENV-infected mesentery (right). Note that the endothelial junction marker, CD31, appears reduced and that NS3 staining is only present in the DENV-infected panel (right). L designates the lumen of the blood vessel in both panels. (**B**) Mice deficient in IFN-α,β,γ (strain AG129)
*Figure 5. Continued on next page*

in DENV2-infected individuals were highest on average in both DF of DHF patients compared to DENV1- and DENV3-infected patients (*Figure 6D*). Although the serum used here was derived from a study where there was no increased incidence of DHF diagnosis in patients with secondary infection over primary infection (*Low et al., 2006*), secondary infection is considered to be a risk factor for increased likelihood of DHF (*Halstead, 2007*). For this reason, we also compared the levels of chymase in these two groups of primary and secondary infection and found that while there were no differences in the chymase levels in patients diagnosed with DF between primary and secondary infection groups (*Figure 6E*), interestingly, significantly higher chymase levels were detectable in DHF patients with secondary infection compared to DHF patients diagnosed during their primary infection (*Figure 6F*). We also observed that for DF, there was no discernable correlation between the amount of virus in the serum and serum chymase levels; however, for DHF patients, there appeared to be a strong correlation between high levels of chymase and lower DENV genome copies in the serum (*Figure 6G*), perhaps, highlighting the potential of MCs to contribute to virus clearance. Human DENV-infection follows a disease course where acute infection is characterized by high viremia and fever, followed by either resolution or hemorrhagic complications that usually occur during the defervescent phase, as viral titers decline (*Halstead, 2007*). These opposing outcomes are difficult to predict early in infection. Our results suggest that, beyond the potential role of chymase in directly promoting vascular events, it may be a useful biomarker for prediction of severe DENV disease outcomes.

## Discussion

Our findings highlight the critical role that MCs may play in virus-induced vascular leakage. Since animals that lack MCs have reduced vascular permeability during DENV infection, to baseline levels, it appears that these cells are highly consequential to the initiation of vasculopathy, which is characteristic of severe DENV cases. TNF has previously been identified to promote the severe vascular leakage that occurs in DENV-infected animals (*Atrasheuskaya et al., 2003*) and our data that leukotrienes and proteases also promote vascular leakage during infection with DENV suggests that multiple mediators, a substantial portion of them MC-derived, likely act in concert to effectuate the vascular complications that occasionally occur during severe DENV infections (*Figure 7*). Since DENV is naturally a primarily human-restricted

*Figure 5. Continued*

were infected with DENV by intra-peritoneally injection of $2 \times 10^5$ PFU of Eden2. After 1 day, treatment was initiated for some infected mice by administering intra-peritoneally injections of cromolyn. On day 3, blood was collected from untreated and cromolyn-treated infected groups and uninfected controls. Hematocrit analysis was performed using blood from individual mice n ≥ 3. Error bars represent the SEM and * indicates a significant increase over uninfected controls and ** indicates a significant decrease compared to DENV infection alone. The p-value for the comparison between uninfected vs DENV + cromolyn was not significant. The graph in the right panel depicts the plaque forming units obtained using pooled serum. Error bars represent the SEM of the assay, which was performed in replicates. Where no error bars are apparent, values obtained were the same for each replicate. * indicates a significant increase for the cromolyn-treated animals compared to infection alone.

pathogen and information gleaned from each animal model of disease has inherent caveats (*St John et al., 2013*), we have verified our findings that MCs are activated during DENV by examining human clinical samples. Surprisingly, for those few patients that progress beyond normal DENV-associated malaise to develop hemorrhagic complications, we observe that heightened responses of MCs are associated with DHF and are apparent in an extreme elevation of serum chymase. If corroborated in independent patient populations and prospective human studies, this connection may be of great clinical utility since severe cases are notoriously difficult to predict early in infection at the time points where elevated chymase was first detected in this study. Patient serum was also derived in this case from adults experiencing DENV1, DENV2, or DENV3 infections and further studies will also be required to extend these results to children, a patient group that frequently experiences DHF (*Halstead, 2007*), and to additional

strains of virus. In support of the potential role of MC proteases during DENV infection, another recent study has also observed elevated chymase and tryptase in the serum of human patients (*Furuta et al., 2012*). Chymase, itself, can promote vascular leakage over a notably prolonged timecourse (*He and Walls, 1998*; *Kunder et al., 2011*) and its presence in the serum would signify that additional vasoactive cytokines and mediators would have been released from MCs, potentially, also contributing to vascular permeability.

Although we have identified MCs as critical cellular catalysts of DENV-induced vascular pathology, in our view, this does not diminish the important role of the virulence mechanisms by which DENV achieves systemic or substantial infection. In some studies, but not others, high viral titers have been associated with DENV disease severity (*Vaughn et al., 2000*). Interestingly, we observe that for DHF patients, a decrease in serum virus genome copies is correlated with increased levels of MC products in the serum. We believe that this observation points to the dual role of MCs in immune protection and pathology. Our findings may provide one mechanistic explanation of how systemic infection may translate into a substantial pro-inflammatory response with negative implications for vascular integrity in terms of its barrier function. Just as many risk factors have been identified for DENV patients that are associated with severe infection and vascular leakage, many factors can theoretically influence the degree of MC activation in vivo in response to a stimulus, including the stimulus concentration, the presence of pre-existing antibodies such as IgE and their concentration, and host-specific differences in the intracellular signaling and sensitization of MC that can result from genetic diversity or existing allergy.

These studies have also identified several candidate drugs that target MCs or their products that may be effective in limiting or preventing DENV pathology in a clinical setting and each has the added advantage of already being approved for human use in other contexts (*McFadden and Gilbert, 1992*; *Leff et al., 1998*). Interestingly, all of these drugs would be expected to limit host immune activation without antiviral activity. Although use of the Sash MC-deficient mouse model suggests that MCs are major contributors of the vasoactive factors that promote vascular permeability during DENV infection, we have used drugs to block MCs and MC products that are not entirely specific to inhibiting MC function. For example, cromolyn can reduce the activation levels of additional cell types (*Holian et al., 1991*). Although leukotrienes were targeted because they are produced by MCs in copious amounts and we show that MC-derived leukotrienes can directly induce permeability of endothelial cell monolayers, they are also produced by cell types in addition to MCs (*Dahlen et al., 1981*). Therefore, the benefits of MC targeting may be in part due to the blockade of inflammatory products from other cellular sources. Our observation that MC stabilization also promotes modestly increased viremia, at least when used in an immunocompromised mouse model where virus replication cannot be effectively controlled, also appears to support the contrasting role of MCs in the effective containment of DENV infection (*St John et al., 2011*), occurring in this case independent of interferon responses. Although

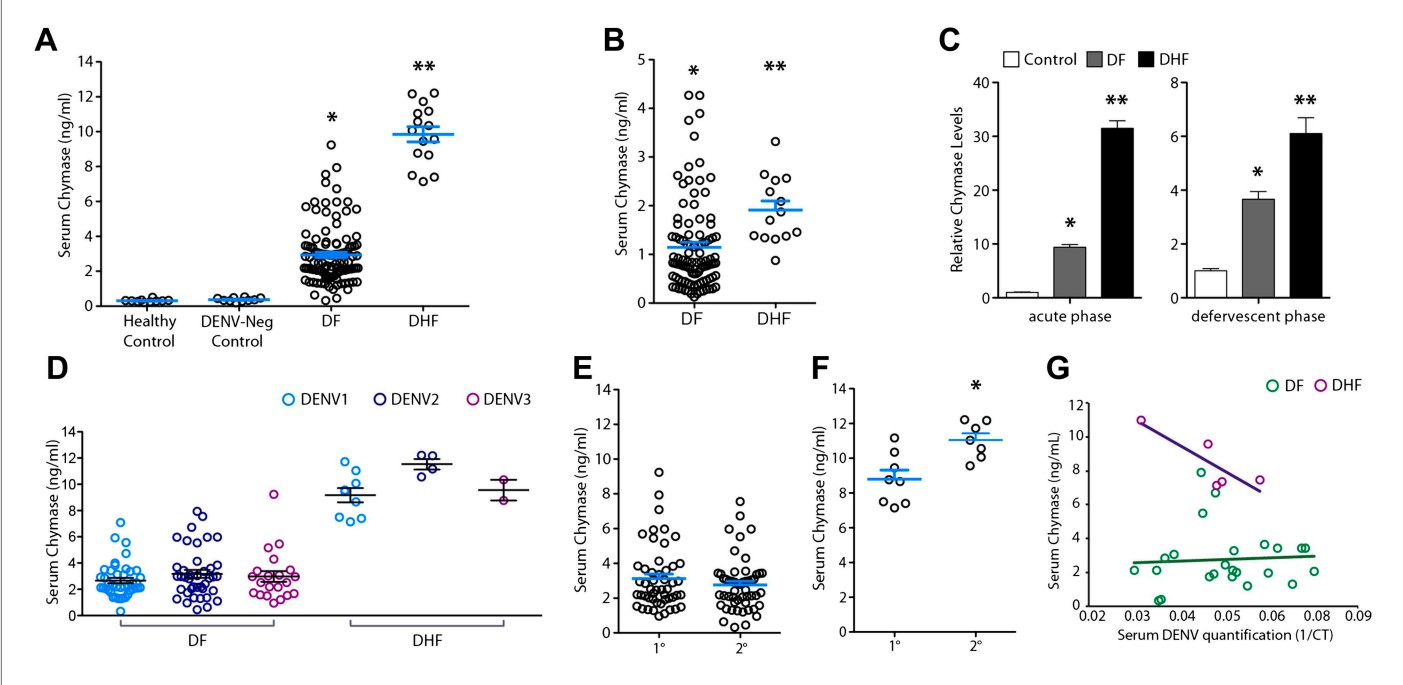

**Figure 6**. Severity of DENV-induced disease in humans is linked to the MC product chymase. (**A**) Graph depicts the chymase concentration in human serum for healthy controls, DENV-negative febrile patients, and patients that were diagnosed with DF or DHF and positive for DENV by molecular tests (see 'Materials and methods'). For DENV-Neg, DF and DHF patients, serum was collected during acute infection, 2–4 days after the onset of fever. (**B**) Graph depicts the serum chymase concentration in DF and DHF patients 4–7 days after fever onset (defervescent phase). For (**A** and **B**), each dot represents the average concentration for an individual patient (n = 10–108 patients per group). (**C**) Data is represented as the relative amount of chymase in patient samples obtained in the acute phase (left) or defervescent phase (right), after normalizing to the average chymase concentration in healthy control human serum. For (**A**–**C**) ANOVA analysis was used to determine significance of samples with Bonferroni's post-test to determine significance between groups. * indicates a significant increase over healthy controls and DENV-Neg, febrile controls and ** indicates a significant increase over healthy control, DENV-Neg control, and DF groups. p<0.0001. (**D**) Graph depicts the concentration of chymase in serum samples grouped based on the serotype of DENV with which the patient was infected. Analysis by two-way ANOVA to compare chymase concentrations amongst DF and DHF samples reveals that serotype significantly influenced the chymase levels in patient sera, p<0.0001, although contributing to only 2.6% of the total variation. The concentrations of chymase in (**E**) DF or (**F**) DHF patients with either primary (1°) or secondary (2°) infection are shown. Chymase levels were significantly higher during secondary infection *p=0.0049 for DHF patients, but did not differ for DF patients, determined by Student's unpaired t-test. (**G**) The concentrations of chymase are plotted vs the corresponding amounts of virus genome copies amplified from serum samples (represented as the inverse of the cross-over threshold [CT] value determined by real time PCR). For DF samples (green), Pearson's R = 0.06, indicating no correlation. For DHF samples (purple), Pearson's R = −0.85, indicating a correlation between higher viral genome copies and lower chymase levels.

The following figure supplements are available for figure 6:

**Figure supplement 1**. Serum chymase levels are presented to represent the repeated measures for individual patients at the early (acute) and late (defervescent) time points of infection and a line connects each patient's paired values.

we do see that viral replication occurs in our WT mouse model over a time course of several days, it is interesting to point out that the mechanism of MC degranulation in response to DENV is not dependent upon intracellular replication of virus (**St John et al., 2011**). Experimental infection of MCs with DENV is possible in vitro and could potentially occur in vivo where it would likely enhance the cytokine responses to infection; however, MCs appear to directly detect the structure of DENV virus particles during degranulation. When this occurs as an early detection event during cutaneous infection, detection of DENV can be protective for the host, for example, by promoting the recruitment of NK cells and NKT cells into a site of infection to aid in viral clearance (**St John et al., 2011**). Widespread MC activation may become pathological when viral titers are high on a systemic level and, as in additional pathological contexts, MC responses in the context of systemic infection could reach a point where they become detrimental to the host. In addition to the evidence that direct sensing of DENV structure occurs by MCs (**St John et al., 2011**), other indirect mechanisms exist for MCs to degranulate in

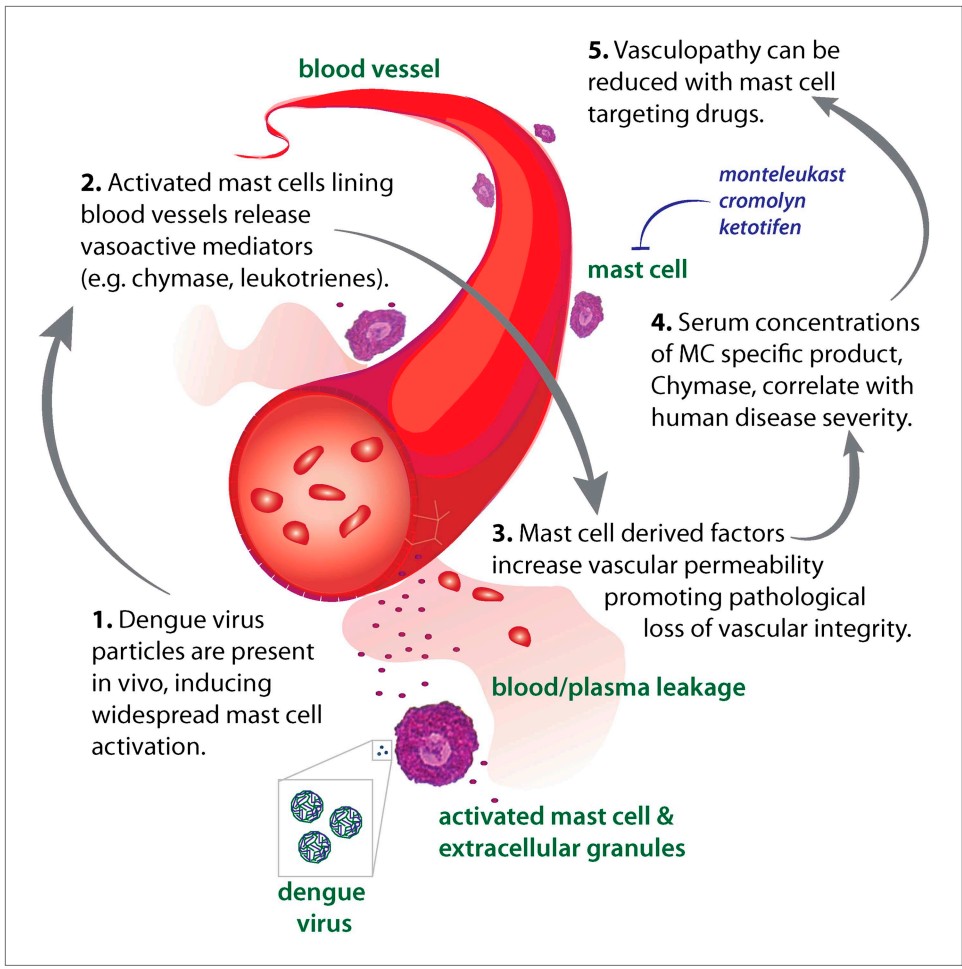

**Figure 7**. Diagram representing the impact of DENV-induced activation of MCs on the vasculature. High DENV viral titers in vivo results in the activation of MCs, which release many vasoactive factors in the vicinity of blood vessels including leukotrienes and proteases, such as chymase. These factors act in concert to promote vascular leakage that, when occurring on a systemic level, has pathological consequences for the host. Drugs that target MC products can limit this leakage and vascular pathology. Similarly, the MC-specific product chymase can also be used to predict the severity of hemorrhagic disease in human patients.

response to pathogens in vivo which could provide additional activating stimuli for MCs during infection. For example, MCs are able to degranulate in response to components of the complement pathway (*McLachlan et al., 2003*) and complement activation has also been associated with DENV severity and plasma leakage in human patients (*Malasit, 1987*).

Many infections are capable of generating events such as cytokine storm in infected hosts; yet, the pathogen-specific manifestations of DENV (including the substantial degree to which vascular leakage can occur) highlights that unique mechanisms underlie the interaction between the virus and host during this infection. MCs appear to have highly specific responses to unique viruses and very few viruses have been identified that are capable of causing direct degranulation of MCs (*Abraham and St John, 2010*). Interestingly, respiratory viruses are primarily thought to not induce direct degranulation of MCs (without antibody sensitization or augmentation of MC responses due to host factors) (*Abraham and St John, 2010*). This is consistent with our data from non-DENV febrile human patients, most of which had been diagnosed as having respiratory viral infections (*Low et al., 2006*), where systemic MC activation was not detected in terms of chymase levels. Another virus that was recently recognized to induce MC-degranulation is vaccinia virus, which is not a hemorrhagic virus and is also primarily a cutaneous and not blood borne virus (*Wang et al., 2012*). These observations point to a DENV-specificity of the MC response that may be dependent on the ability of virulent DENV to reach

systemic titers and elevate MC products in the blood. This may have further implications in the context of related and unrelated viral hemorrhagic fevers and suggests that further investigations are warranted into the role of MCs in hemorrhaging induced by additional viral pathogens.

Within the context of treatment for DENV vasculopathy, the dual role of MCs in protection and pathology also emphasizes the importance of carefully testing any therapeutic regimen targeting MCs in human patients and perhaps considering a combination therapy approach that also would limit viral replication when such a strategy is available. Since those individuals with the highest levels of MC products during this study also experienced the most clinically severe disease, defined by vascular leakage or hemorrhaging, our data also highlight the possibility of using MC products as biomarkers to identify individuals at highest risk for hemorrhagic complications. Investigating the mechanism of DENV-induced degranulation may also reveal additional targets for blocking MC activation by DENV. For additional pathogens, such as *E. coli*, unique receptors on MCs have been identified to promote MC activation. Identification of receptors and signaling components involved in the direct activation of MCs by DENV is likely to impact our understanding of both DENV protection and pathology.

## Materials and methods

### Animal studies

MC-deficient mice ($W^{sh}/W^{sh}$; "Sash"), *Alox5*-deficient (5-LO-KO), TNF-KO, and control mice (+/+; wild type, on a C57BL/6 background), were originally purchased from Jackson Laboratories. Additional C57BL/6 mice were purchased from the National Cancer Institute. Sash mice were repleted with in vitro matured bone marrow–derived MCs from congenic controls to generate reconstituted Sash (Sash-R) mice as previously described (*St John et al., 2011*). Sv/129 mice deficient in type I and II IFN receptors (strain AG129, denoted as IFN-$\alpha$,$\beta$,$\gamma$-R$^{-/-}$) were originally purchased from B&K Universal and provided as a gift by Dr Subhash Vasudevan. All experiments were performed according to protocols approved by the DUKE-NUS or Duke University Division of Laboratory Animal Resources and their respective University Institutional Animal Care and Use Committees.

### Infections and drug treatments

For infections, DENV serotype 2, strain Eden2, was used. This clinical isolate was derived from the same study (Early Dengue Infection and Outcomes Study, Eden ; *Low et al., 2006*) as the patient sera in *Figure 6*. The culture conditions we used to maintain this strain and its interactions with MCs were previously described (*St John et al., 2011*). Mouse infections were performed by intra-peritoneally injection of $1 \times 10^6$ PFU (WT and Sash mice) or $2 \times 10^5$ PFU (IFN-$\alpha$,$\beta$,$\gamma$-R$^{-/-}$ mice). Cromolyn (3 mg/mouse/day) and ketotifen (0.6 mg/mouse/day) (both from Sigma) were injected intra-peritoneally in PBS as vehicle either 1 hr (WT and Sash mice) or 24 hr (IFN-$\alpha$,$\beta$,$\gamma$-R$^{-/-}$ mice) after infection. Montelukast (brand Singulair; Merck) was administered (0.4 mg/mouse) by oral gavage 1 hr after infection by crushing tablets using a mortar and pestle and resuspending in PBS.

### Vascular studies

Vascular leakage was assessed at the reported time points after infection using an Evans blue leakage assay. Evans blue was prepared as a 1% solution in PBS, syringe filtered and 100 μl of this solution was injected by tail vein, 30 min prior to euthanasia. Immediately following death, animals were perfused with 15 ml of PBS (approximately fivefold the mouse blood volume) per animal by opening the chest cavity, creating an incision in the right atrium of the heart and injecting the full PBS volume in the left ventricle of the heart. Blood was, thus, flushed fully from the mouse circulatory system. Tissue biopsies were then obtained, their mass recorded, and homogenized in PBS. Cellular debris was cleared by centrifugation, and the OD-600 was measured to quantify the amount of dye within tissues, based on a standard curve. These values were normalized to the tissue mass. For hematocrit analysis, heparinized blood that had been collected from individual mice via the maxillary vein was run on an automatic hematology analyzer to obtain values. ELISAs for MCPT1 were performed using a kit from R&D Systems, according to manufacturer's instructions.

### Immunofluorescence microscopy

Spleen sections were frozen in OCT compound (Tissue-Tek), and frozen sectioned (10 μm thick) using a cryostat (Leica). Sections were acetone-fixed at 4°C, then blocked with PBS containing 1% BSA prior to staining with primary antibodies against anti-CD11b, anti-CD11c (eBiosciences) and dsRNA (clone J2; English and Scientific Consulting Bt.). The following secondary antibodies were then used:

anti-mouse-AlexaFluor555, streptavidin-AlexaFluor488, and anti-rat-Cy5 (Invitrogen). Slides were then mounted and imaged by confocal microscopy. For whole mounting, mouse mesentery was dried onto slides, fixed with formaldehyde, and permeabilized/blocked using 0.1% saponin (Sigma) in PBS containing 1% BSA. Primary staining was then performed in permeabilization/blocking buffer at 4°C, overnight, using a rat antibody against CD31 (eBioscience), a rabbit antibody against DENV NS3 (GeneTex), and the MC-specific probe, Avidin conjugated to FITC (Sigma). After extensive washing with PBS, tissues were incubated overnight with secondary antibodies: AlexaFluor 660-conjugated anti-rat (Molecular Probes), and Cy3-conjugated anti-rabbit (Jackson Immunoresearch). Images were obtained under epifluorescence.

## Human clinical samples

Human serum samples were derived from a bank of samples from adult febrile or healthy control individuals in Singapore (*Low et al., 2006*; *Fink et al., 2007*). Samples used for this study were derived from patients that ranged in age from 18 to 77 years, with a mean of 40 years; 42% of samples were obtained from females, 58% from males. Dengue positive samples were determined based on physician diagnosis as well as molecular tests including RT-PCR viral RNA. The DF and DHF patient sera used in this study were previously determined to be positive for serotypes 1, 2, or 3 by RT-PCR. Of the samples included in this study, approximately 47% of the DHF cases and 50% of the DF cases were secondary infections, which had been determined previously by testing for dengue-specific IgM and IgG (*Low et al., 2006*). Non-DENV positive patients were febrile patients that were referred to the original study as suspected DF patients, but were negative for confirmatory DENV markers by molecular tests. Data is derived from sera obtained at the time of enrollment on days 1–3 after the start of fever and from a paired sample that was obtained days 4–7 after the start of fever. The patient's physician made DHF versus DF diagnosis during the original study, based on WHO 1997 guidelines. These diagnoses have also been given the designation of 'severe' (DHF) and 'mild' (DF) within the Eden study. All patients supplied written informed consent. The methods of the original study from which these samples were obtained, which was approved by the National Healthcare Group IRB, are included as references *Low et al. (2006)* and *Fink et al. (2007)*. ELISA kits for Human chymase were obtained from Antibodies-Online and performed according to manufacturer's instructions. Relative chymase levels in the serum displayed in *Figure 6C* were calculated by normalizing the concentration in each sample to the average concentration of chymase in healthy control serum donors. To quantify viral genome copies, viral RNA was isolated from 100 μl of serum using the QIAamp Viral RNA Mini Kit (Qiagen), followed by amplification and detection using the Qiagen OneStep RT-PCR Kit, both used according to manufacturers instructions. We used serotype-specific primers for DENV1-3 designed by the CDC that were described previously (*Chien et al., 2006*). A subset of samples was used for PCR detection of virus and this selection was based solely on having sufficient remaining serum to perform the assay.

## Transwell assays

Mouse vascular endothelial cell lines, EOMA and SVEC4-10EHR1, were obtained from ATCC through the Duke University Tissue Culture Repository. Cells were grown on transwells with 3-μm pores (BD Biosciences) for 4 days for a monolayer to form. To isolate MCs, the peritoneal and pleural cavities were flushed twice with PBS containing 0.5% BSA and 2 mM EDTA. Cells were then washed and resuspended in anti-c-kit primary antibody (eBioscience) at a concentration of 5 μg/ml and incubated for 1 hr at 4°C. After washing, the MACS method was used to purify MCs by incubating labeled cells with anti-rat-IgG magnetic beads prior to isolation using LS columns (both from Miltenyi), according to manufacturer's instructions. Some purified cells were cytospun onto slides and stained to visualize MC morphology. Isolated cells were resuspended in culture medium and either incubated alone, or treated with DENV at an MOI of 5 for 1 hr at 37°C in a tissue culture incubator. Media containing DENV alone was incubated concurrently for use as a control. To isolate MC supernatants, after incubation, cells were centrifuged to pellet them and a volume equivalent to supernatant from $5 \times 10^5$ cells per well) was applied gently to the trans-well insert for a final volume of 400 μl. The TER of the endothelial monolayers was measured at baseline, prior to treatment and at 12 or 24 hr after exposure.

For trans-well assays using BMMCs, an MOI of 1 was used and drugs were used to inhibit the actions of MCs or their products on the endothelial cells. For cromolyn treatment, cromolyn (10 μM; Sigma) was incubated with BMMCs during DENV activation. Immediately prior to exposing cells to MC ± DENV supernatants, inhibitors for leukotrienes (montelukast, 10 μM final concentration; Sigma) or

chymase (Soybean Trypsin Inhibitor, 100 µM final concentration and Chymostatin, 30 µM final concentration; both from Sigma) were added to trans-wells. Supernatants were then added to each insert.

## Data analysis

Images were prepared for publication using ImageJ. Prism 5 and Excel were used to determine statistical significance. For direct comparisons of infected and control samples, Student's un-paired t-test was used. For comparisons of multiple groups, ANOVA was performed with Bonferroni's post-test to determine statistical significance. Two-way ANOVAs were used in the case of assessing data for multiple groups at multiple time points. Data were considered significant at $p \leq 0.05$.

## Acknowledgements

We thank Duane Gubler and Eng Eong Ooi for their discussions. Cheryl Chan is thanked for performing Sash mouse reconstitutions and Subhash Vasudevan is thanked for providing AG129 mice. We also thank Darvi Sergio, Bryan Buenaflor, and Yohannes Asfaw for assistance performing hematocrit readings and Gladys Ang for critical manuscript review. We acknowledge the Eden study and thank Yee Sin Leo and Eng Eong Ooi for the human sera.

## Additional information

### Funding

| Funder | Grant reference number | Author |
|---|---|---|
| New Investigator Award by the Singapore National Medical Research Council (NMRC) | NMRC/NIG/1054/2011 | Ashley L St John |
| National Institutes of Health | | Soman N Abraham |
| Singapore National Medical Research Council (NMRC) | NMRC/TCR/005-NUS/2008 | Mah-Lee Ng |
| Duke-NUS Signature Research Program funded by the Agency for Science, Technology and Research (A*STAR), Singapore, and the Ministry of Health, Singapore | | Soman N Abraham |

The funders had no role in study design, data collection and interpretation, or the decision to submit the work for publication.

### Author contributions

ALS, Project conception, Experimental design, Performed experiments, Analysis and interpretation of data, Wrote the manuscript; APSR, Project conception, Experimental design, Analysis and interpretation of data, Reviewed the manuscript; BR, Performed ELISAs using human clinical samples, Reviewed the manuscript; M-LN, Provided human clinical samples, Reviewed the manuscript; SNA, Project conception, Data interpretation, Reviewed the manuscript

### Ethics

Human subjects: All patients supplied written informed consent. The original study from which these samples were obtained was approved by the National Healthcare Group Institutional Review Board (identifiers DSRB B/05/013 and DSRB B/09/432). *Low et al. (2006)* and *Fink et al. (2007)* contain detailed reports of the methods of the original study.

Animal experimentation: All experiments were performed according to protocols approved by the DUKE-NUS (Protocol #SHS/710) or Duke University Division of Laboratory Animal Resources (Protocol #A318-10-12) and their respective University Institutional Animal Care and Use Committees.

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
