## [Decision Letter]

Thank you for choosing to send your work entitled “Involvement of Mast Cells and Their Vasoactive Products, Leukotrienes and Chymase, in Dengue Virus-Induced Hemorrhaging” for consideration at *eLife*. Your article has been favorably evaluated by a Senior editor and 3 reviewers, one of whom is a member of our Board of Reviewing Editors.

The Reviewing editor and the other reviewers discussed their comments before we reached this decision, and the Reviewing editor has assembled the following comments to help you prepare a revised submission.

This study provides the evidence that activation of mast cells by DENV contributes to vascular leakage. This finding is clinically important and can inform future developments of therapy for DENV infections.

However, there are some concerns related to the authors’ interpretation of the data derived from the human dengue cohort and in several instances the authors seem to over-interpret their data. One major concern is the authors’ use of the word “hemorrhage” when “vascular leakage” would be more appropriate. Another concern stems from their patient studies – demographics should be described in more detail and results of their chymase assays should be presented differently (Figure 6). Lastly, the Materials and methods section should be more detailed and should include methods that are currently described in the Results and Figure legends.

Major comments:

1) The authors inappropriately use the terms “vascular leakage” and “hemorrhage” interchangeably throughout the manuscript. Although these two clinical signs are hallmarks of DENV infection, according to the 1997 WHO criteria by which diagnosis of patients whose blood samples were included for analysis in this study was made, only vascular or plasma leakage, but not hemorrhage, is specific to DHF as it is the only clinical setting that differentiates severe DHF from mild DF. A portion of DF cases can also develop signs and symptoms of hemorrhage such as ecchymosis, and bleeding from gum or nose. In most instances, hemorrhage can only be diagnosed from clinical signs and symptoms such as melena and hematemesis, which are associated with gastrointestinal bleeding, petechiae (skin bleeding), or gum or nose bleeding. There is however, in some DHF cases with severe hemorrhage, lower hematocrit values, especially following an episode of rising hematocrit. Therefore, changes in the two parameters observed in WT animals infected by DENV described in this study i.e., a rise in hematocrit and an extravasation of albumin into tissue as assessed by quantification of an albumin-binding dye, Evans blue, likely indicate increased vascular permeability or plasma leakage in response to DENV infection, but not hemorrhage as stated in several places by the authors. Hemorrhage should only be definitely defined clinically or histologically, which was not described anywhere in the manuscript. This point must be clarified and corrected, including within the title.

2) Although the authors refer to previous publications (Early Dengue infection and outcome study (EDEN) – study design and preliminary findings, *Ann Acad Med Singapore* 35 (11), 783-789 [2006] and Host gene expression profiling of dengue virus infection in cell lines and patients, *PLoS Negl Trop Dis* 1 (2), e86 [2007]), demographics of the cohort, especially for the cases with clinical samples used in this particular study, should be described in more detail and discussed in the paper (e.g.,, age, infecting DENV serotypes, primary or secondary infections) as these parameters have been shown to influence the outcome of disease. As referred to in Early Dengue infection and outcome study (EDEN) – study design and preliminary findings, *Ann Acad Med Singapore* 35 (11), 783-789 (2006), the blood specimens used in this study were obtained from the adult dengue cohort in Singapore. Epidemiologically, DENV infection in adults has different clinical outcomes compared to children: it tends to develop milder symptoms and has a lower risk of developing DHF, which may be due to distinct pathologic mechanisms. In several endemic countries, DENV infection primarily affects children, despite the fact that a number of adult cases have been increasingly observed. It is therefore worth mentioning and discussing in the paper.

3) In several places in the manuscript (including the Abstract and a Results subheading), the authors claim that the MC-specific product, chymase, is a predictive biomarker for DHF, as well as MC activation predicting hemorrhage in DENV infection (the latter statement is confusing, as discussed in point number 1). It seems to be too early and too much of an over-interpretation to make such a conclusion with limited data as presented in Figure 6. There are several problems with this dataset, and the way in which the data are displayed and analyzed:

• 3A: As the clinical signs and symptoms of DENV infection dynamically change over time, especially days/hours before defervescence, it would be more informative to display serum chymase levels (raw data not fold changes) in each group of patients relative to days before and after the defervescent date, instead of combining the results from days 2–4 (acute phase) vs days 5–7 (referred to as defervescent phase in the paper).

• 3B: Related to 3A, since duration of the febrile phase in each patient varies greatly (2–7 days), it would be more accurate and more relevant to analyze the data of blood chymase levels based on each disease day. This would give a better picture of the possibility of using it as a predictive biomarker for severe DENV infection.

• 3C: In order to make a statement that blood chymase levels can predict the development of DHF at early phase of the disease, additional statistical methods should also be applied with the dataset (not just analysis of the mean): e.g., a method that accounts for repeated measures over time in a given individual, comparing between groups of control, DF and DHF patients.

• 3D: Figure 6B: the data should be presented as individual chymase blood levels in each patient rather than the average of fold changes, which could be misleading. Also, it is not clear how fold change was calculated.

• 3E: What is/are the point/s to make from the data of Figure 6C that show the difference between chymase levels in primary DHF versus secondary DHF? Does mast cell activation occur more in secondary DHF? If so, why? Is it because of higher viremia in those cases, thereby triggering more mast cell activation and degranulation? If yes, the viral load in those patients should also be presented. These points need to be discussed.

• 3F: To support the authors’ hypothesis, it would be more informative to show the data of hematocrit and platelet numbers in those patients along with chymase levels.

4) Is dengue viremia observed in the animal model established in this study?

5) Although previous studies suggest the association of TNF with vascular leakage, assessment of vascular permeability in this study by TNF blocking and TER in TNF-KO mice did not show a significant role of TNF. Why is that? The authors should explain or discuss this in detail.

6) In this study, cromolyn was found to increase virus titer in serum of immunocompromised (IFN-R^-/-^) mice. Therefore, it is possible that this drug can reduce vascular pathology but on the other hand may promote virus replication. The authors should demonstrate whether or not cromolyn and other MC stabilizers have any effect on virus titer in serum or tissues of the established immunocompetent mice model.

7) This study provides possible drug treatments to reduce vascular pathology in DENV infection. The data is derived from in vivo experiments using mouse models of DENV infection. Medications were given to infected animals during early time points after infection (30 min for immunocompetent or 1 day for immunocompromised mice). Taking into consideration also the data from the authors’ previous study, mast cell degranulation seems to have both protective and pathogenic roles. If this is also true in DENV infection in humans, administration of drugs that affect this process must be carried out very carefully: not too early but early enough to prevent detrimental inflammatory reaction caused by MC activation products. This point may be worth discussing.

---

## [Author Response]

*1) The authors inappropriately use the terms “vascular leakage” and “hemorrhage” interchangeably throughout the manuscript. Although these two clinical signs are hallmarks of DENV infection, according to the 1997 WHO criteria by which diagnosis of patients whose blood samples were included for analysis in this study was made, only vascular or plasma leakage, but not hemorrhage, is specific to DHF as it is the only clinical setting that differentiates severe DHF from mild DF. A portion of DF cases can also develop signs and symptoms of hemorrhage such as ecchymosis, and bleeding from gum or nose. In most instances, hemorrhage can only be diagnosed from clinical signs and symptoms such as melena and hematemesis, which are associated with gastrointestinal bleeding, petechiae (skin bleeding), or gum or nose bleeding. There is however, in some DHF cases with severe hemorrhage, lower hematocrit values, especially following an episode of rising hematocrit. Therefore, changes in the two parameters observed in WT animals infected by DENV described in this study i.e., a rise in hematocrit and an extravasation of albumin into tissue as assessed by quantification of an albumin-binding dye, Evans blue, likely indicate increased vascular permeability or plasma leakage in response to DENV infection, but not hemorrhage as stated in several places by the authors. Hemorrhage should only be definitely defined clinically or histologically, which was not described anywhere in the manuscript. This point must be clarified and corrected, including within the title*.

We appreciate the reviewers' clear argument regarding the use of the word hemorrhage. While we have not eliminated the word “hemorrhaging” entirely from the manuscript, it is now confined to the background sections and to the diagnosis of dengue hemorrhagic fever in patients, since this is the appropriate clinical diagnosis term. In general, when discussing our own mechanistic data obtained in mouse models we have changed to the wording “vascular leakage”.

*2) Although the authors refer to previous publications (Early Dengue infection and outcome study (EDEN) – study design and preliminary findings, Ann Acad Med Singapore 35 (11), 783-789 [2006] and Host gene expression profiling of dengue virus infection in cell lines and patients,* PLoS Negl Trop Dis 1 *(2), e86 [2007]), demographics of the cohort, especially for the cases with clinical samples used in this particular study, should be described in more detail and discussed in the paper (e.g.,, age, infecting DENV serotypes, primary or secondary infections) as these parameters have been shown to influence the outcome of disease. As referred to in Early Dengue infection and outcome study (EDEN) – study design and preliminary findings,* Ann Acad Med Singapore *35 (11), 783-789 (2006), the blood specimens used in this study were obtained from the adult dengue cohort in Singapore. Epidemiologically, DENV infection in adults has different clinical outcomes compared to children: it tends to develop milder symptoms and has a lower risk of developing DHF, which may be due to distinct pathologic mechanisms. In several endemic countries, DENV infection primarily affects children, despite the fact that a number of adult cases have been increasingly observed. It is therefore worth mentioning and discussing in the paper*.

We have added more information regarding the demographics of the patient cohort from which these samples are derived. We also added the reviewers’ point to the discussion that children frequently experience severe dengue in developing countries and therefore it would be important in future studies to compare chymase levels in adults and children during mild and severe disease.

*3) In several places in the manuscript (including the Abstract and a Results subheading), the authors claim that the MC-specific product, chymase, is a predictive biomarker for DHF, as well as MC activation predicting hemorrhage in DENV infection (the latter statement is confusing, as discussed in point number 1). It seems to be too early and too much of an over-interpretation to make such a conclusion with limited data as presented in Figure 6*.

We have changed the subheading and Abstract to remove the claim that the levels are “predictive” and we have tempered this claim in the Discussion to emphasize that further prospective human studies should be undertaken to determine if chymase can predict DHF with clinical diagnostic sensitivity.

*There are several problems with this dataset, and the way in which the data are displayed and analyzed*:

• *3A: As the clinical signs and symptoms of DENV infection dynamically change over time, especially days/hours before defervescence, it would be more informative to display serum chymase levels (raw data not fold changes) in each group of patients relative to days before and after the defervescent date, instead of combining the results from days 2–4 (acute phase) versus days 5–7 (referred to as defervescent phase in the paper)*.

We cannot change the way the serum was collected and associated data was recorded at this stage since we are using an existing bank of human dengue patient serum. However, we are not clear how the date of fever subsiding would be linked to mast cell activation or mast cell-mediated immune pathology. Showing data relative to the date that fever subsides also does not seem to be common in the existing dengue literature. Therefore, we see fever as another factor, the onset and end of which may coincide with certain stages of DENV infection, but its role in dengue vascular pathology is not clearly described. These time ranges are used to provide information regarding the time that passed subsequent to the development of clinical presentation, which is partially defined by fever onset.

Since chymase levels are significantly elevated at both early and late time points, any changes to the presentation of the data relative to various time points or milestones in the progression of disease would not change the conclusions of the study that chymase was significantly elevated in DF and DHF patients throughout the duration of the study, and beginning with the earliest serum samples obtained.

• *3B: Related to 3A, since duration of the febrile phase in each patient varies greatly (2–7 days), it would be more accurate and more relevant to analyze the data of blood chymase levels based on each disease day. This would give a better picture of the possibility of using it as a predictive biomarker for severe DENV infection*.

This is a good idea and we hope we and/or others can extend upon this initial observation by performing a prospective study to generate a daily time course of chymase levels in dengue patients.

As stated above, while we agree this would provide finer detail of the accumulated MC products each day and when peak levels actually occur, it is important to note that chymase was highly elevated in the initial serum sample obtained from patients at the time of enrollment.

• *3C: In order to make a statement that blood chymase levels can predict the development of DHF at early phase of the disease, additional statistical methods should also be applied with the dataset (not just analysis of the mean): e.g., a method that accounts for repeated measures over time in a given individual, comparing between groups of control, DF and DHF patients*.

This information has been included as Figure 6–figure supplement 1.

• *3D: Figure 6B: the data should be presented as individual chymase blood levels in each patient rather than the average of fold changes, which could be misleading. Also, it is not clear how fold change was calculated*.

The individual chymase blood levels are displayed in Figure 6A. Figure 6C (previously Figure 6B) is included to illustrate the relative amounts of chymase in the serum of patients compared to controls, which are approximately 3-times higher for DF patients and 30-times higher in DHF patients compared to healthy controls. We believe that this is an informative and not misleading point, particularly since the data is represented both ways. However, we have changed the y-axis to read “relative chymase concentration” for clarification and included an explanation of the calculation in the methods. This data is normalized to the serum chymase concentration in healthy individuals.

• *3E: What is/are the point/s to make from the data of Figure 6C that show the difference between chymase levels in primary DHF versus secondary DHF? Does mast cell activation occur more in secondary DHF? If so, why? Is it because of higher viremia in those cases, thereby triggering more mast cell activation and degranulation? If yes, the viral load in those patients should also be presented. These points need to be discussed*.

We have now provided data showing the viral burden in a subset of infected patients and discussed this data further. We believe that these results that chymase levels are higher during secondary infection suggest that mast cell activation may occur more during secondary infection in this cohort of patients. It is important to note that it was previously reported that the patients in this study were not more likely to develop DHF with secondary infection [1]. Heightened mast cell activation could be due to dose-dependent responses to antigen (in this case, virus) or due to other mechanisms resulting in sensitization of mast cells (such as the presence of antigen-specific IgE), or other mechanisms prompting mast cell activation. Interestingly, rather than observing high virus titers concurrently with high levels of mast cell activation, we see the opposite. For DHF patients but not DF patients, heightened mast cell products in the serum are strongly correlated with a decline in detectable virus genome copies in the serum. This could point to the dual role mast cells play in both immune protection and immune pathology. Perhaps the enhanced activation of mast cells reduces the virus titer but also contributes to vascular leakage in DHF patients. This point has been discussed in the main text, as suggested by the reviewers.

• *3F: To support the authors’ hypothesis, it would be more informative to show the data of hematocrit and platelet numbers in those patients along with chymase levels*.

The hematocrit and platelet numbers for these patients have previously been reported [1]; however, we did not find a correlation between either of these values and the chymase levels. Since dengue patients are treated with fluid replacement therapy if they require it, hematocrit values are not usually a reliable measure of vascular leakage in patients that are given the standard of care therapy for dengue virus infection. Although thrombocytopenia, or reduced platelet counts, accompanies dengue, we have focused here on the pathophysiological phenomenon of vascular leakage. Further studies will be required to assess whether additional symptoms of dengue infection such as the lowered platelet counts are influenced by mast cell activation.

*4) Is dengue viremia observed in the animal model established in this study*?

The wild type mouse model is not considered to be a viremia model for dengue virus and we also did not detect virus by plaque forming assay using serum from wild type mice after i.p. infection by the Eden2 strain. However, sera from these mice were transiently PCR positive for dengue virus between 12 and 48 hours after infection. The AG129 mouse model was used in Figure 5 in order to replicate our findings in a more widely accepted dengue viremia model.

*5) Although previous studies suggest the association of TNF with vascular leakage, assessment of vascular permeability in this study by TNF blocking and TER in TNF-KO mice did not show a significant role of TNF. Why is that? The authors should explain or discuss this in detail*.

Previous studies that examined the role of TNF were performed in immunocompromised mice that are unable to restrict the replication of dengue virus. It is possible that in that system the role of TNF and virus replication is enhanced. This may indicate that TNF is a particularly important mediator if the host does not adequately contain viral replication. Certainly, humans experience a more prolonged infection than WT mice and it is possible that different mediators contribute to varying extents along the time course of infection.

*6) In this study, cromolyn was found to increase virus titer in serum of immunocompromised (IFN-R*^*-/-*^*) mice. Therefore, it is possible that this drug can reduce vascular pathology but on the other hand may promote virus replication. The authors should demonstrate whether or not cromolyn and other MC stabilizers have any effect on virus titer in serum or tissues of the established immunocompetent mice model*.

We observed a minor but not statistically significant increase in the amount of virus in the serum with cromolyn treatment of wild type mice. Based on this trend and the data in Figure 5b in the context of immunocompromised mice, it may be that mast cells still have a role in immune protection from dengue virus with i.p. inoculation but, conversely, it also appears that mast cell stabilization does not have widespread detrimental immunosuppressive effects. The limitations to their contributions in dengue clearance in mice could potentially be because we have given a systemic infection, bypassing the stages of progression where MCs are important for peripheral immunosurveillence and, thus, bypassing many of their functions. Since there is a slight increase in pfu equivalents that is not significant, a potential remains that at certain inoculating doses or time points, a significant increase in titer would be perceivable even in wild type mice that are immunocompetent. The dual role in protection and pathology for mast cells revealed in both mouse models here and in our previous studies [3] has implications for human infection as well, as discussed in response to the next question.

*7) This study provides possible drug treatments to reduce vascular pathology in DENV infection. The data is derived from* in vivo *experiments using mouse models of DENV infection. Medications were given to infected animals during early time points after infection (30 minutes for immunocompetent or 1 day for immunocompromised mice). Taking into consideration also the data from the authors’ previous study, mast cell degranulation seems to have both protective and pathogenic roles. If this is also true in DENV infection in humans, administration of drugs that affect this process must be carried out very carefully: not too early but early enough to prevent detrimental inflammatory reaction caused by MC activation products. This point may be worth discussing*.

This is an important point. In the context of human infection, the incubation period for dengue virus between inoculation and the onset of clinical symptoms is around 3 days, minimally, with an average of just under 6 days [2]. We have shown previously that mast cells are key to the initial immune-surveillance for dengue virus, which should occur in the skin just subsequent to the injection of virus particles [3]. Mast cells are also important for promoting the initiation of adaptive immune responses in local draining lymph nodes [4]. Based on our understanding of the kinetics of localized inflammatory and adaptive immune responses, these localized mast cell-accelerated processes that limit early replication and promote adaptive immune responses should have been initiated even before the onset of symptoms that occurs ≥3 days after exposure. Increasingly, we understand that localized responses of mast cells to pathogens are protective, but pathological mast cell activation can result from widespread or systemic triggering of mast cells. We agree that treatment too early with mast cell stabilizers could be detrimental and it might be counter-active to test mast cell stabilizers as a preventative treatment before dengue diagnosis or with possible exposure. We would hypothesize that testing of mast cell activators at the time of dengue diagnosis, which is generally after the onset of fever and after the development of systemic infection, would be a promising strategy to limit overwhelming systemic activation of mast cells while allowing the early and protective mast cell responses to occur without hindrance.

References:

1. Low, J.G., Ooi, E.E., Tolfvenstam, T., Leo, Y.S., Hibberd, M.L., Ng, L.C., Lai, Y.L., Yap, G.S., Li, C.S., Vasudevan, S.G., & Ong, A., Early Dengue infection and outcome study (EDEN) – study design and preliminary findings. *Ann Acad Med Singapore* 35 (11), 783-789 (2006).

2. Chan, M. & Johansson, M.A., The incubation periods of dengue viruses. *PLoS ONE* 7 (11), e50972 (2012).

3. St John, A.L., Rathore, A.P., Yap, H., Ng, M.L., Metcalfe, D.D., Vasudevan, S.G., & Abraham, S.N., Immune surveillance by mast cells during dengue infection promotes natural killer (NK) and NKT-cell recruitment and viral clearance. *Proc Natl Acad Sci U S A* 108 (22), 9190-9195 (2011).

4. Abraham, S.N. & St John, A.L., Mast cell-orchestrated immunity to pathogens. *Nat Rev Immunol* 10 (6), 440-452 (2010).